# Sheet Metal Forming Optimization Methodology for Servo Press Process Control Improvement

**Antonio Del Prete and Teresa Primo \***

Department of Engineering for Innovation, University of Salento, Via per Arnesano, Building "O", 73100 Lecce, Italy; antonio.delprete@unisalento.it

**\*** Correspondence: teresa.primo@unisalento.it

**Abstract:** In sheet metal forming manufacturing operations the use of servo presses is gaining more interest due to the opportunity to improve process performance (quality, productivity, cost reduction, etc.). It is not yet clear how to proceed in the engineering process when this type of operating machine is used to achieve the maximum possible potential of this technology. Recently, several press builders have developed gap- and straight-sided metal forming presses adopting the mechanical servo-drive technology. The mechanical servo-drive press offers the flexibility of a hydraulic press with the speed, accuracy and reliability of a mechanical press. Servo drive presses give the opportunity to improve the productivity of process conditions and improve the quality of stamped parts. Forming simulation and numerical optimization can be useful tools to define beforehand the optimal process parameter set-up in terms of servo press downward curve properties. This is done by carrying out a sensitivity analysis of the forming parameters having influence on said curve. The authors have developed a numerical methodology able to analyze the influence factors, for comparison with the degrees of freedom made available by the usage of a servo press, in terms of stroke profile management, to obtain an optimized process parameters combination.

**Keywords:** servo press; metal forming; automotive; optimization

## 1. Introduction

Electro-mechanical servo-drives have been used in machine tools for several decades. Recently, several press builders, mainly in Japan [1,2] and Germany [3], developed metal forming presses able to utilize the mechanical servo-drive technology. The mechanical servo-drive press offers the flexibility of a hydraulic press (infinite sliding-ram speed and position control, availability of press force at any slide position) with the speed, accuracy and reliability of a mechanical press [4]. The advantage of the servo motor is that it can control all the press motions, such as speed, stroke, slide motion and position. The mechanical servo press flywheel, clutch and brake are replaced by high-capacity motors, and thus the maintenance of the servo press is simplified. Compared with traditional presses that have been used for many years, the mechanical servo press allows users to obtain higher productivity, better product quality, simpler set up and maintenance, and high repeatability. One of the most important advantages of the servo press is the flexible slide movement [5]. As discussed above, the following advantages arise from choosing a motion suitable for each aim.

1. Accumulated 'know-how' obtained from use of existing, traditional presses can be inherited because motions such as crank press and linkage press can be duplicated by a servo press.
2. Impact loading is avoided, and the tool's life is extended by reducing the contact speed when the tool hits the blank.

3. Lubrication is often improved and the working limit can be extended by using a pulsating or oscillating slide motion.
4. Contact and break-through noise is reduced by stopping the slide for a short time or reducing the slide speed.
5. Blank vibration can be reduced by an optimized slide motion so the shape of sheet meetal product is stabilized.
6. The product quality can be improved by controlling the slide parallelism and choosing an optimum slide motion.
7. Higher productivity is possible by shortening of a forming cycle, with a partial short stroke around bottom dead center as well as a high-speed return motion.

Wrinkles represent one of the most frequent defects during deep drawing processes that should be avoided; the function of the blank holder force is to effectively suppress wrinkle formation. Conversely, a high blank holder force increases the frictional force, which tends to cause blank ruptures. Thus it is important to reduce the friction to achieve a successful deep drawing operation. Tamai et al. [6] tried to improve the formability of high-strength steel parts in deep drawing by detaching the tools from the die cushion periodically from the sheet. It was found that the sheet was automatically re-lubricated when the tool was detached. Komatsu and Murakami [7] reported a case where wrinkling in deep drawing was prevented by applying the stepwise drawing motion on a servo press with a constant clamping force. Wrinkles were eliminated by applying a smaller (about 1/3) blank holder force than that of the conventional motion with a stepwise motion. A similar effect for preventing the occurrence of wrinkling by the pulsating internal pressure was reported for tube hydroforming [8]. A sheet product, which ruptures under the conventional crank motion, is successfully formed by optimizing the slide motion of a servo press [9]. In this motion, the punch touches the sheet at a slow speed, and the slide movement is reversed once between the pre-forming stage of the top portion and the drawing stage of the rectangular portion.

The development of the servo press method is accelerating as the capacity of servo motors becomes bigger. In the future, it is expected that this method will replace conventional press methods, thereby improving product quality, increasing productivity, maintaining tools integrity, and reducing energy consumption. Motion control in the servo press method has to be effectively optimized depending on the shape and material characteristics. However, in the industrial field, the motion control settings relied on the experience or intuition of the most-skilled workers. Workers could not avoid having to undertake much trial and error to find the optimized motion law [10]. Chanhee et al. [11,12] carried out experimental validations by applying the design variables suggested by the safe forming window for the multi-stage forming.

Wei et al. [13] performed numerical analyses with a multi-objective genetic algorithm for optimizing BHF (Blank Holder Force) and draw beads simultaneously to minimize fracture and wrinkling during the deep drawing process.

In this paper, the authors report an attempt to develop a technique to optimize the servo press motion law to obtain the maximum process benefit.

## 2. Numerical Methodology Description

To achieve the study objectives, the authors developed a numerical methodology which allows analysis of the influence factors for comparison with the degrees of freedom made available by the servo press, in terms of stroke profile management.

Specifically, the study followed the following steps:

1. Development, implementation and simulation of a numerical plan applied to an industrial test case, considering the actual material used online—a low carbon steel (DC04 steel sheets, commonly used in the automotive industry [14]);
2. Analysis of obtained results;

3. Numerical data correlation;
4. Optimization model implementation;
5. Analysis and performance evaluation of the obtained results compared with the developed optimization model;
6. Definition and resolution of the optimization problem for two different blank geometries;
7. Optimized model validation.

### 2.1. Development, Implementation and Simulation of a Numerical Plan Applied to an Industrial Test Case

The chosen reference test case is a component for an automotive application, a wheel fender, obtained through a Cx2 processing method (from a single blank it is possible to obtain two parts: the right and left fenders). Based on the current industrial process, the finite element (FE) model of numerical simulation was calibrated to the real model.

Finite element analysis (FEA) was used to understand the deformation behavior of a material during the forming process. In this paper, the commercial finite element code Radioss®was used to run explicit forming simulations. HyperForm®was used to create the finite element mesh, assign the boundary conditions and to build Radioss input deck for the analysis. The punch, die and the blank holder were created using rigid materials, while Yoshida–Uemori material was used for the blank. For the forming analysis we used shell elements and, to reduce the calculation time while maintaining accuracy, an adaptive meshing scheme was used. The FE model of the tooling and the blank size are shown in Figures 1 and 2.

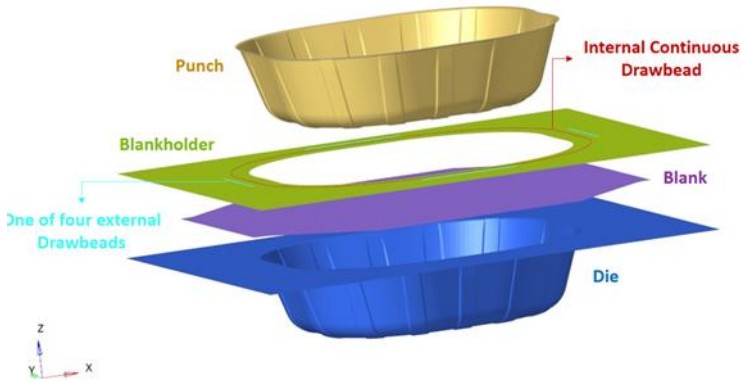

**Figure 1.** Finite element (FE) model created for simulation of the sheet forming process.

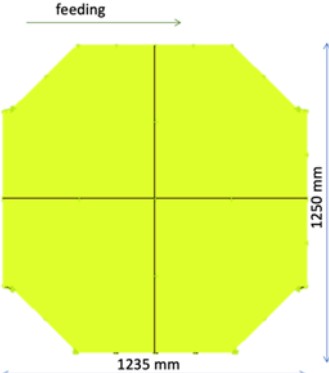

**Figure 2.** Blank size.

Table 1 reports sheet material properties used for the blank and the friction coefficient used in the simulation [15,16]. The friction between the blank and the tool parts was modelled by Coulomb's law.

**Table 1.** DC04 (low-carbon steel) material properties and friction coefficient used in the simulation.

| $\sigma y$ (MPa) | $Rm$ (MPa) | $\nu$ | $h$ | $B_0$ (MPa) | $C$ | $m$ | $B$ (MPa) | $R_{sat}$ (MPa) | $n$ | $\mu$ |
|---|---|---|---|---|---|---|---|---|---|---|
| 130 | 500 | 0.3 | 0.526 | 168 | 657.9 | 1.281 | 8.980 | 558.6 | 0.19 | 0.125 |

In particular:

$\sigma y \rightarrow$ Yield stress

$\nu \rightarrow$ Poisson's ratio

$h \rightarrow$ Material parameter for controlling work hardening stagnation

$m \rightarrow$ Parameter for isotropic and kinematic hardening of the bounding surface

$b \rightarrow$ Center of the bounding surface

$\mu \rightarrow$ Coulomb friction

$Rm \rightarrow$ ultimate stress

$B_0 \rightarrow$ Initial size of the bounding surface

$C \rightarrow$ Parameter for kinematic hardening rule of yield surface

$R_{sat} \rightarrow$ Saturated value of the isotropic hardening stress

$n \rightarrow$ hardening coefficient

The simulation of the traditional process refers to a constant punch speed equal to 2000 mm/s (this data set-up is referred to as the simulation context). This run has been indicated as RUN0 and represents the traditional drawing operation. The draw beads have been numerically modeled as a geometric profile made by a line to which the analytical (friction) and geometric (shape) parameters are associated. Figures 3 and 4 show the results of the forming process simulation for RUN0. These outputs represent the basis for response extrapolation used for the analytical optimization model. Failure criteria used in the analysis were based on the forming limit diagrams (FLD). Specifically, in Figure 3, FLD is shown, considering the initial blank with maximum dimensions equal to 1235 mm x 1250 mm and the blank nominal thickness equal to 0.77 mm.

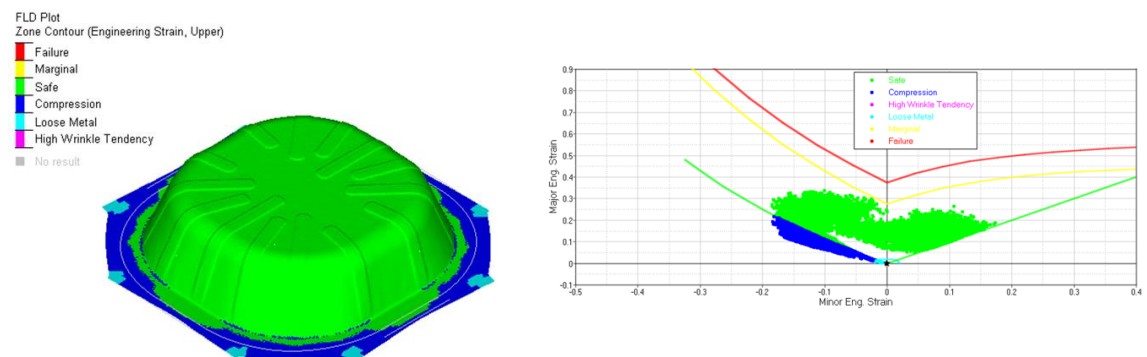

**Figure 3.** Forming limit diagram (FLD) for RUN0.

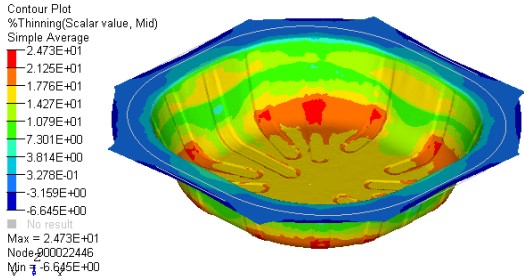

**Figure 4.** % Thinning distribution for RUN0.

In Figure 4, the percentage-thinning map is shown. It is possible to observe how the maximum thinning, mainly corresponding to the fillet radii at the end of the stroke, is equal about to 25%.

Figure 5 shows the comparison between the product obtained by forming with a traditional mechanical press and the numerical model. Compared to what is usually evaluated in output, i.e., the maximum and minimum percentage thinning and maximum reaction forces, the model response has been also evaluated with respect to the flow of formed material in the die for industrial interest.

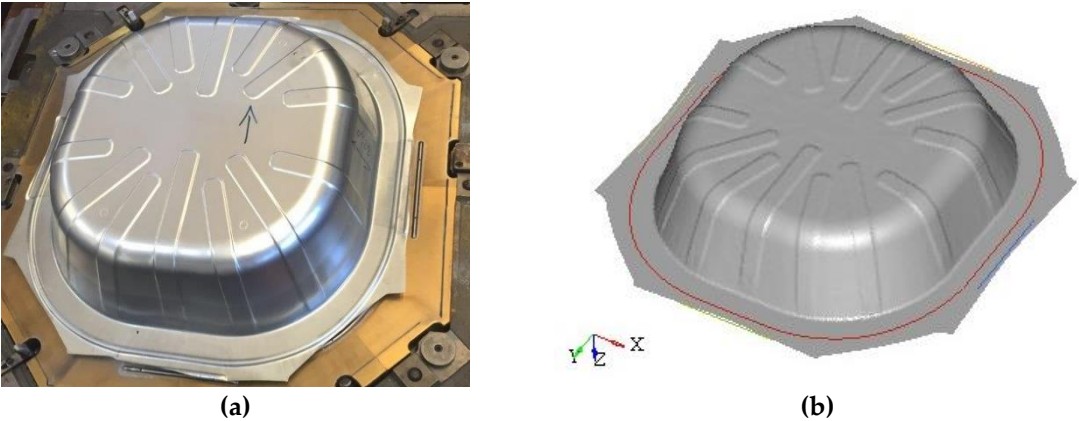

| (a) | (b) |

**Figure 5.** Comparison between: (**a**) the real product obtained by forming using a traditional mechanical press and (**b**) the numerical model.

Once the output variables X1, X2, Y1 and Y2 have been defined, the maximum distance of the linear edges of the blank has been reported from the first draw bead border (Figure 6). The output variables set is shown in Table 2.

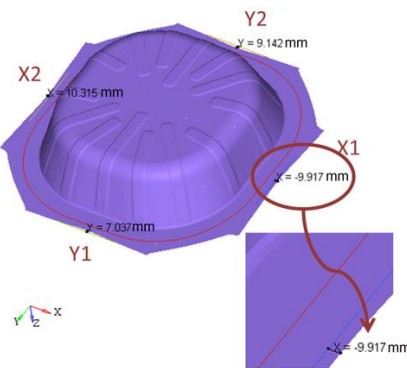

**Figure 6.** Definition of output variables: X1, X2, Y1 and Y2 and related detail of X1.

**Table 2.** Output variables list.

| Max Thickness (%) | Min Thickness (%) | Max Rforce DIE (N) | X1 (mm) | X2 (mm) | Y1 (mm) | Y2 (mm) |
|---|---|---|---|---|---|---|
| Output 1 | Output 2 | Output 3 | Output 4 | Output 5 | Output 6 | Output 7 |

Outputs 4, 5, 6 and 7 are considered negative if the blank exceeds the position of the draw bead with respect to which the measure is taken (in the sense of the formed material flow in the die), and considered positive if, instead, the blank does not exceed the draw bead edge.

The introduction of kinematic control with servo presses allows users to better take advantage of the intrinsic characteristics of the material with respect to the deformation state. The material behavior has a dependency on the strain rate, but it mainly has a dependency on the deformation state. In particular, the material changes its performance based on its deformation history. This behavior is the main feature that can be exploited through a driven process with a servo press. Starting from these considerations, the authors have developed a procedure that allows investigation of precisely this

behavior, introducing, not only a change in the slide speed, but also a disengagement interval of the forming tool from the blank during forming. This type of slide stroke is called "stepwise", because the stroke of the slide has return steps of the slide itself.

Figures 7 and 8 show details of the input variables for the definition of the experiment plan, through which it is possible to identify the input variables of the defined simulation plan. For the specific case of interest only one disengagement has been considered.

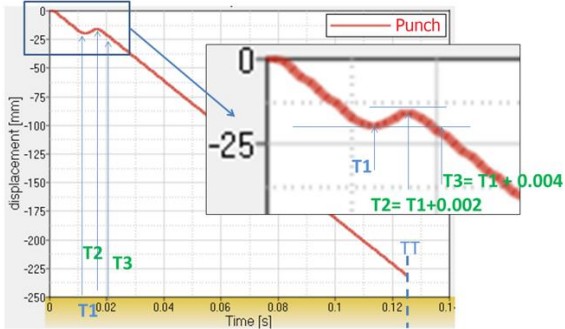

**Figure 7.** Input timing definition on stepwise curve.

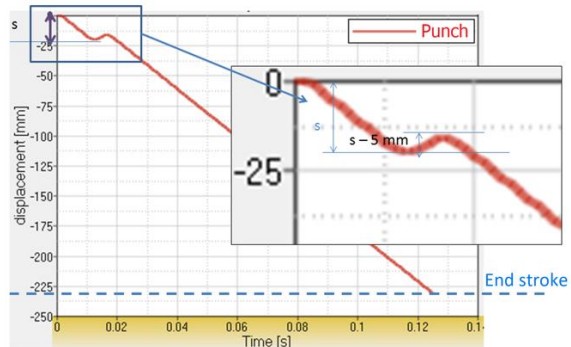

**Figure 8.** Input stroke definition on stepwise curve.

In particular, the input variables are:

- S → tool stroke point where the return takes place
- T1 → time at which the return occurs (speed inversion)
- T2 → return end time (velocity inversion) with position equal to S + 5 mm
- T3 → return to S position
- TT → termination time, coinciding to a stroke equal to "end stroke".

In the case of RUN0, T1 coincides with TT, while S coincides with the total travel of the tool. For each of the identified input variables the following constraints have been defined:

- 50 mm < S < FF − 10 mm
- 0.009 s < T1 < 0.01 s
- 0.046 s < TT < 0.233 s
- FF = 231 mm (fixed value)
- T2 = T1 + 0.002 s (value fixed by T1)
- T3 = T1 + 0.004 s (value fixed by T1).

Where FF is the end stroke, T2 and T3 are variables depending on T1. The runs, related to the defined experimental plan, characterized by the variables described above, are reported in Table 3.

**Table 3.** Design variables for experiment plan runs.

| Run# | S (mm) | T1 (s) | TT (s) | Run# | S (mm) | T1 (s) | TT (s) |
|------|--------|--------|--------|------|--------|--------|--------|
| 1 | 50 | 0.0748 | 0.1874 | 14 | 162.64 | 0.0804 | 0.1681 |
| 2 | 99.28 | 0.0916 | 0.1729 | 15 | 155.6 | 0.0972 | 0.2068 |
| 3 | 211.92 | 0.0944 | 0.1439 | 16 | 183.76 | 0.0496 | 0.1148 |
| 4 | 57.04 | 0.0468 | 0.1826 | 17 | 141.52 | 0.1 | 0.139 |
| 5 | 226 | 0.0832 | 0.202 | 18 | 85.2 | 0.0636 | 0.1536 |
| 6 | 134.48 | 0.044 | 0.2165 | 19 | 64.08 | 0.0412 | 0.1342 |
| 7 | 197.84 | 0.0356 | 0.2116 | 20 | 190.8 | 0.0608 | 0.1923 |
| 8 | 78.16 | 0.086 | 0.1245 | 21 | 169.68 | 0.0776 | 0.1197 |
| 9 | 204.88 | 0.0384 | 0.1632 | 22 | 106.32 | 0.058 | 0.11 |
| 10 | 218.96 | 0.0664 | 0.1487 | 23 | 127.44 | 0.0328 | 0.1294 |
| 11 | 92.24 | 0.0888 | 0.2213 | 24 | 148.56 | 0.0524 | 0.1584 |
| 12 | 176.72 | 0.072 | 0.231 | 25 | 71.12 | 0.0552 | 0.2262 |
| 13 | 120.4 | 0.0692 | 0.1971 | 26 | 113.36 | 0.03 | 0.1778 |

Based on Table 3, for each input file, starting from RUN0, the tool displacement curve and the termination time of the simulation have been modified. The other two points of the curve included between T1 and TT are the points T2 and T3, where relative displacement S − 5 mm and again S, are automatically calculated. The plan described above relates to a single blank geometry (blank 0). In reality, to better evaluate the advantage with respect to material savings, a second plan has been evaluated, identical to the first, but for modified blank dimensions (blank 1). A size reduction in the perpendicular direction to the blank feed, equal to 7.5 mm, has been defined. This choice derives from the possibility of always being able to use the same tool for shearing, but with a smaller width coil. A blank of smaller dimensions is thus obtained without any change for the shearing tool.

*2.2. Discussion of the Obtained Results*

From the numerical analysis set up described above two simulation plans have been obtained. The results, for the outputs described in the Table 2, are reported below for the blank 0 (Table 4), and for the blank 1 in (Table 5).

The maximum thickness reduction is obtained in RUN7, while the maximum thickness increase value (wrinkles probability) is obtained in RUN1. The maximum and minimum reaction force values are obtained, respectively, for RUN8 and RUN16/20, while the maximum material recall in the die occurs for the RUN4, RUN11 and RUN25.

For the plan relating to blank 1, the maximum and minimum percentage of thinning values were obtained, respectively, for RUN9 and RUN6. The maximum and minimum reaction force values were obtained, respectively, for RUN5 and RUN17, while the maximum material recall in the die occurred for RUN1, RUN25 and RUN26. The results of the analysis show that the blank dimension reduction has an influence on the outputs. The results related to the geometric variable must be analyzed considering the value of X1 and X2. For the two blanks they are always different because the blank size changes in the X direction. It is therefore an output that cannot be considered in an absolute sense, but it is strongly dependent on the blank geometry. The post-processing of some of the simulations carried out for the simulation plan is reported below. In particular, Figures 9 and 10, show the percentage-thinning distribution and X1, X2, Y1 and Y2 values for RUN20 and RUN22 for the blank0, while Figures 11 and 12 show the same output for the blank1 for RUN1 and RUN11. For all the highlighted runs it is evident how, when a stepwise happens, it is possible to detect a reaction force reduction (Figures 13–16).

**Table 4.** Output of BLANK 0 simulation plan.

| Run | Max Thickness (%) | Min Thickness (%) | Max Rforce DIE (N) | X1 (mm) | X2 (mm) | Y1 (mm) | Y2 (mm) |
|---|---|---|---|---|---|---|---|
| 1 | 20.2 | −7.5 | $2.02 \times 10^6$ | 0.632 | −0.517 | −16.984 | −18.473 |
| 2 | 20.0 | −7.3 | $1.92 \times 10^6$ | 0.508 | −0.048 | −16.191 | −18.851 |
| 3 | 22.1 | −6.9 | $2.01 \times 10^6$ | 5.97 | 5.691 | −10.963 | −12.754 |
| 4 | 19.3 | −7.4 | $1.97 \times 10^6$ | −0.996 | −0.523 | −17.991 | −19.98 |
| 5 | 22.1 | −7.1 | $1.99 \times 10^6$ | 5.058 | 5.365 | −11.243 | −13.195 |
| 6 | 21.0 | −7 | $2.01 \times 10^6$ | 3.536 | 3.576 | −13.534 | −15.507 |
| 7 | 25.3 | −6.3 | $2.00 \times 10^6$ | 14.672 | 14.819 | −3.418 | −5.069 |
| 8 | 23.1 | −6.6 | $2.76 \times 10^6$ | 9.776 | 9.201 | −8.404 | −10.098 |
| 9 | 25.0 | −6.2 | $1.98 \times 10^6$ | 14.01 | 14.455 | −3.736 | −5.168 |
| 10 | 22.6 | −6.7 | $2.01 \times 10^6$ | 6.975 | 7.556 | −9.517 | −11.036 |
| 11 | 20 | −7.4 | $2.04 \times 10^6$ | −0.839 | 0.209 | −18.533 | −19.615 |
| 12 | 21.6 | −7 | $2.02 \times 10^6$ | 4.415 | 4.591 | −12.61 | −14.069 |
| 13 | 19.6 | −7.3 | $1.98 \times 10^6$ | 1.049 | 1.192 | −15.7 | −17.665 |
| 14 | 21.1 | −7 | $1.93 \times 10^6$ | 3.787 | 3.828 | −13.306 | −15.226 |
| 15 | 20.7 | −7.2 | $1.89 \times 10^6$ | 3.298 | 3.819 | −13.848 | −15.51 |
| 16 | 23.5 | −6.7 | $1.20 \times 10^6$ | 10.651 | 10.94 | −6.804 | −8.568 |
| 17 | 21.8 | −6.9 | $2.00 \times 10^6$ | 6.657 | 6.285 | −10.66 | −12.524 |
| 18 | 19.3 | −7.4 | $2.13 \times 10^6$ | −0.483 | 0.4 | −16.86 | −18.779 |
| 19 | 19.7 | −7.4 | $2.10 \times 10^6$ | −0.513 | 0.362 | −16.986 | −19.325 |
| 20 | 23.3 | −6.9 | $1.20 \times 10^6$ | 6.344 | 6.501 | −10.528 | −12.538 |
| 21 | 22.8 | −6.9 | $1.98 \times 10^6$ | 7.009 | 6.67 | −10.403 | −11.965 |
| 22 | 20.6 | −7 | $1.87 \times 10^6$ | 4.996 | 4.54 | −12.106 | −14.586 |
| 23 | 21.6 | −7 | $1.89 \times 10^6$ | 5.988 | 5.698 | −11.908 | −13.041 |
| 24 | 21.5 | −7 | $2.05 \times 10^6$ | 5.272 | 4.765 | −12.078 | −13.765 |
| 25 | 19.1 | −7.4 | $−2.02 \times 10^6$ | −0.675 | −0.919 | −17.808 | −19.328 |
| 26 | 20.3 | −7.3 | $1.95 \times 10^6$ | 2.953 | 2.545 | −14.286 | −16.651 |

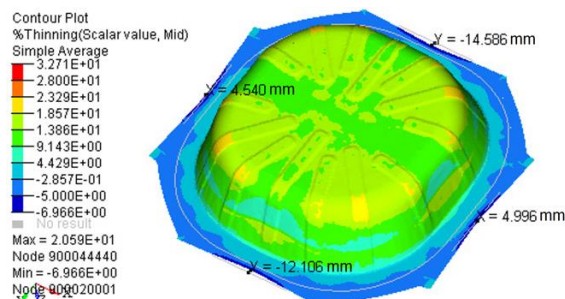

**Figure 9.** % Thinning distribution and X1, X2, Y1 and Y2 values for the RUN22; blank0.

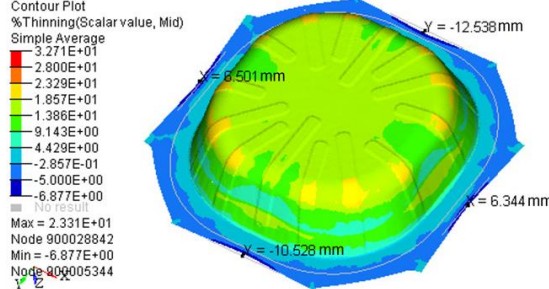

**Figure 10.** % Thinning distribution and X1, X2, Y1 and Y2 values for the RUN20; blank0.

**Table 5.** Output of BLANK 1 simulation plan.

| Run | Max Thickness (%) | Min Thickness (%) | Max Rforce DIE (N) | X1 (mm) | X2 (mm) | Y1 (mm) | Y2 (mm) |
|---|---|---|---|---|---|---|---|
| 1 | 20.2 | −7.5 | $1.84 \times 10^6$ | −8.1 | −7.7 | −15.3 | −15.5 |
| 2 | 19.5 | −7.6 | $1.96 \times 10^6$ | −4.7 | −4.4 | −18 | −16.9 |
| 3 | 21.6 | −7 | $1.98 \times 10^6$ | −2.1 | −1.9 | −11.2 | −10.5 |
| 4 | 19.1 | −7.3 | $1.87 \times 10^6$ | −6 | −5.9 | −18.9 | −19.3 |
| 5 | 21.9 | −7.3 | $2.37 \times 10^6$ | −1.9 | −1.2 | −11.2 | −12.1 |
| 6 | 21.9 | −8.2 | $2.05 \times 10^6$ | −7.5 | −7.8 | −25 | −25.8 |
| 7 | 24 | −7.5 | $2.17 \times 10^6$ | −2.9 | −3.6 | −15.7 | −14.4 |
| 8 | 22.8 | −6.6 | $1.88 \times 10^6$ | −3.4 | −3.4 | −4.4 | −4.1 |
| 9 | 24.3 | −7.2 | $2.18 \times 10^6$ | −7 | −7.6 | −12.3 | −13.6 |
| 10 | 24.2 | −7.7 | $2.06 \times 10^6$ | −2.5 | −2.8 | −17.5 | −18.4 |
| 11 | 19.1 | −7.5 | $1.94 \times 10^6$ | −8.5 | −8.2 | −20.3 | −19.9 |
| 12 | 21.9 | −7.2 | $2.04 \times 10^6$ | 0.3 | 0.3 | −13.6 | −17.2 |
| 13 | 19.4 | −7.4 | $1.86 \times 10^6$ | −5.8 | −5.5 | −16.8 | −14.9 |
| 14 | 21.2 | −7 | $2.00 \times 10^6$ | −2.7 | −1.7 | −14.2 | −12.8 |
| 15 | 20.1 | −7.3 | $2.09 \times 10^6$ | 3.9 | 4.7 | −15.7 | −14.9 |
| 16 | 23.4 | −7.5 | $2.08 \times 10^6$ | 0.2 | 0.7 | −17.5 | 18.5 |
| 17 | 22.3 | −7 | $1.75 \times 10^6$ | 0.5 | 0.6 | −10.7 | −9.9 |
| 18 | 19.8 | −7.6 | $2.06 \times 10^6$ | −6 | −5.7 | −18.6 | −17.2 |
| 19 | 21.2 | −7.6 | $1.82 \times 10^6$ | −7.1 | −6.9 | −18.6 | −17.3 |
| 20 | 22 | −7.8 | $2.02 \times 10^6$ | −3.5 | −3.4 | −20 | −20.6 |
| 21 | 22.7 | −6.8 | $1.95 \times 10^6$ | 1.4 | 1.5 | −10 | −9.5 |
| 22 | 21.9 | −7.2 | $2.07 \times 10^6$ | −0.9 | −0.5 | −12.2 | −11.8 |
| 23 | 21.2 | −7.8 | $1.91 \times 10^6$ | 0.7 | 1 | −20.5 | −21.4 |
| 24 | 20.9 | −8.1 | $2.10 \times 10^6$ | −5.7 | −5.7 | −23.4 | −24.9 |
| 25 | 19.1 | −7.6 | $1.91 \times 10^6$ | −8.1 | −7.9 | −19.5 | −18 |
| 26 | 19.3 | −8.1 | $1.95 \times 10^6$ | −5 | −4.9 | −25 | −26.4 |

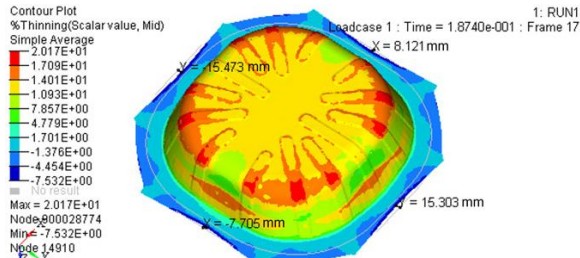

**Figure 11.** % Thinning distribution and X1, X2, Y1 and Y2 values for the RUN1; blank1.

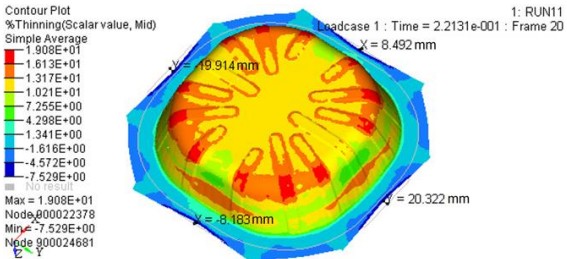

**Figure 12.** % Thinning distribution and X1, X2, Y1 and Y2 values for the RUN11; blank1.

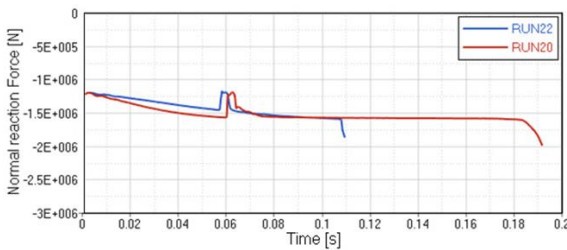

**Figure 13.** Reaction force curves comparison between RUN20 and RUN22; blank0.

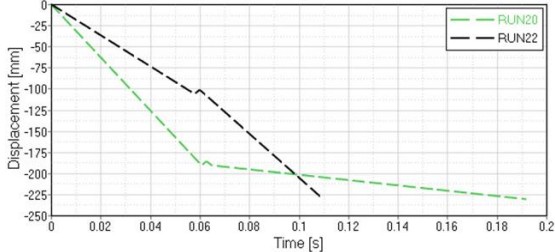

**Figure 14.** Punch displacement with stepwise, in correspondence of S and T1. TT represents the last point of the curve with maximum value in the x axis; blank0.

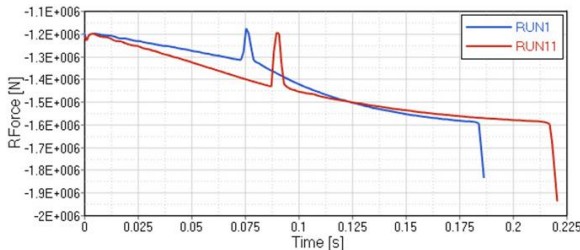

**Figure 15.** Reaction force curves comparison between RUN1 and RUN11; blank1.

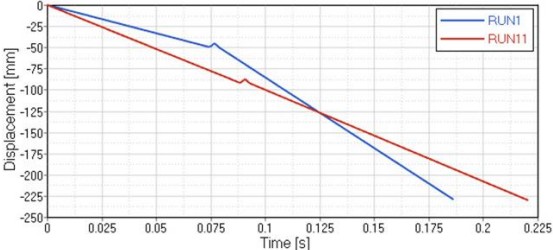

**Figure 16.** Punch displacement with stepwise, in correspondence of S and T1. TT represents the last point of the curve with maximum value in x axis; blank1.

The output values X1, X2, Y1 and Y2 are considered negative if they exceed the draw bead profile (on the 4 sides) in the recall direction of material in the die. They will be positive only in the case of material excess compared to the draw bead profile. All four of the above cases present a slope variation of the reaction force in the coining phase. Furthermore, all considered cases report possible feasibility conditions, because the maximum thinning value, indicated as critical for material rupture equal to 28%, is never exceeded.

## 3. Optimization Model Implementation

Starting from the obtained results, shown in Tables 4 and 5, it has been possible to proceed with the implementation of the optimization model. The model has been investigated for all the output variables that have been identified and divided into two phases: in the first one, the blank shape has been considered as an input variable, defined as "complete plan"; and, in the second that instead

analyzes two separate and distinct plans, defined as "semiplans", one for each blank (blank0 and blank1), then the blank is not considered as an input variable. The complete plan is reported in the present paper. In particular, an optimization procedure has been developed by the integration of the optimization tool Dassault Systèmes ISight with Altair Radioss®solver. The reduction of the maximum reaction force has been chosen as the objective function of the optimization phase. The optimal set-up in terms of process variables definition has been investigated using multi-island genetic algorithm (MIGA) optimization algorithm.

### 3.1. Main Effect Analysis for Complete Plan

The descriptive graphics of the main effect (main effect plot, MEP) have been used to examine the differences between the average levels of the response of interest, for one or more factors (process variables). There is a "main effect" when different levels of a factor influence the response differently. A main effect plot represents, for each level of the considered factors, the response averages connected by a line. These graphs provide indications of the factors that have the greatest influence on the process response variability. For the maximum thickness percentage reduction (max thick% output), the MEP (Figure 17) shows that the blank size does not have the same influence as it appears to have in the case of TT, T and S. This consideration is evident from how much the outputs deviate from the average value. In the particular case of the S variable, there is a growing response trend with respect to the max thick% output. The effect on the minimum thickness percentage reduction (min thick%) output variable is, on the other hand, less evident than the maximum percentage reduction (max thick%). The graph in Figure 18 shows, in fact, values very close to the average. Furthermore, the slope of the values obtained for the blank size variable is always relative to an interval very close to the average value. In contrast, the output of the reaction force has a low influence on all the input variables; if some points are excluded (anomalies) it can be seen (Figure 19) that the responses are very close to the average value. The possible changes of the blank geometry particularly affect the X1 and X2 output values. As shown in Figures 20 and 21, the identified input variables show a considerable influence on the response values represented in these MEP, and, in particular, blank size one strongly influences these outputs.

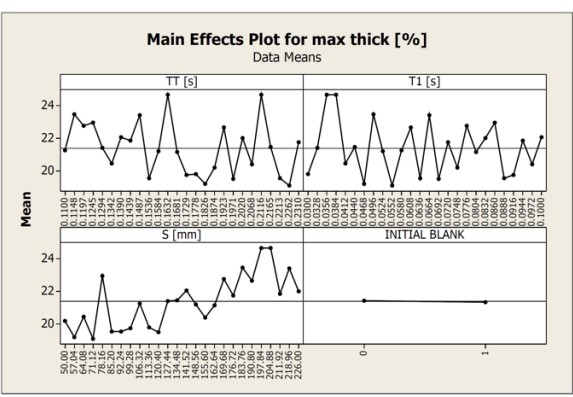

**Figure 17.** MEP for max thick % output.

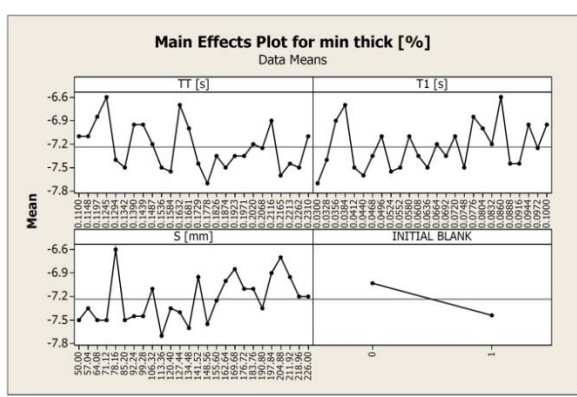

**Figure 18.** MEP for min thick % output.

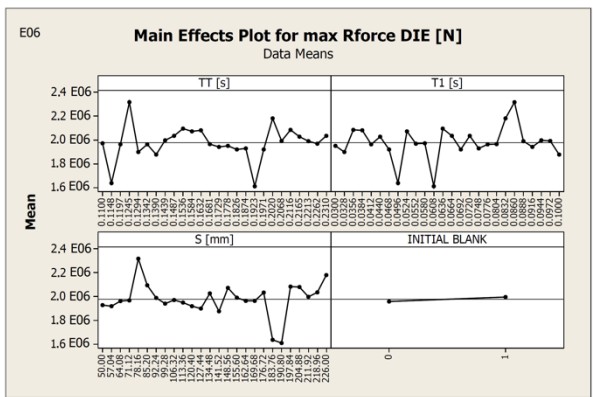

**Figure 19.** MEP for reaction force output.

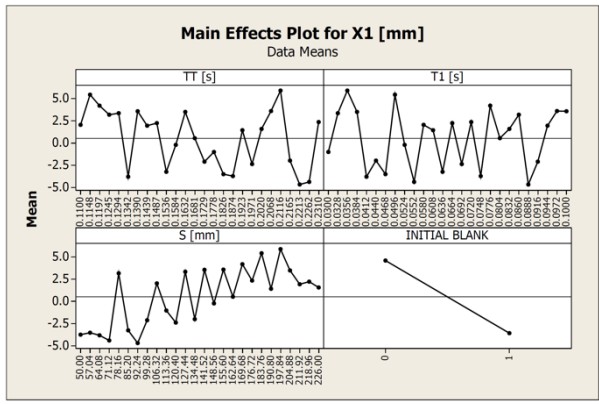

**Figure 20.** MEP for X1 output.

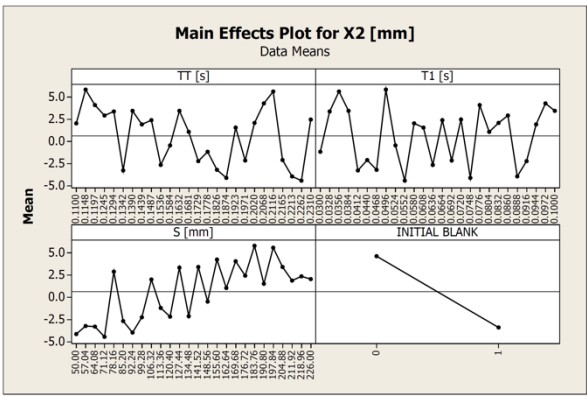

**Figure 21.** MEP for X2 output.

Regarding the outputs Y1 and Y2, as shown in Figures 22 and 23, the identified input variables (TT, T1 and S) show an influence on the output values while the blank size effect is drastically reduced. The Y dimension is not altered in the transition between the blank0 and blank1.

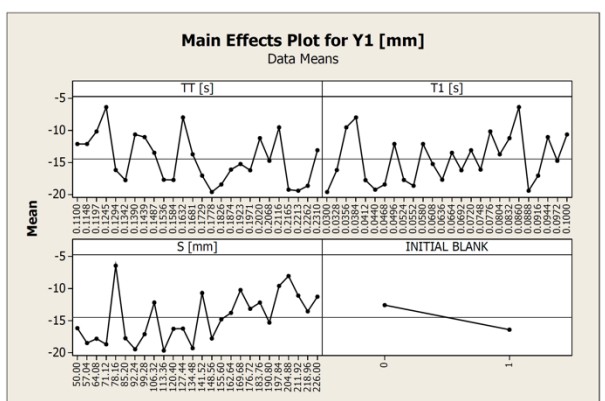

**Figure 22.** MEP for Y1 output.

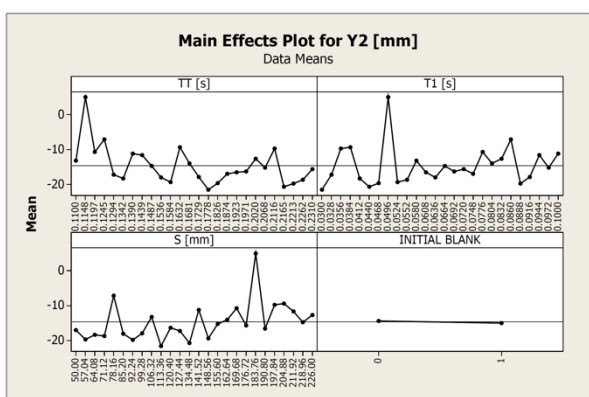

**Figure 23.** MEP for Y2 output.

From the previous graphs, it can be seen that, in general, the TT, T1 and S parameters significantly influence the response variability with respect to its average value. The blank size variable, on the other hand, does not have a significant influence on the considered response variability.

### 3.2. Definition and Resolution of the Optimization Problem for Blank0

In this specific case, three process parameters have been taken into consideration as possible variables: tool stroke S (mm), speed inversion time T1 (s) and termination time TT (s).

The optimization problem has been defined as follows:

| | |
|---|---|
| Objective Function: | Min (max_Rforce_Die) |
| Design Variables: | 50 mm ≤ S ≤ 226 mm, |
| | 0.03 s ≤ T1 ≤ 0.1 s, |
| | 0.11 s ≤ TT ≤ 0.231 s |
| Constraints: | 19% ≤ max_thick ≤ 28%, −8% ≤ min_thick ≤ −5% |

The design space constraints have been assigned by the maximum and minimum values of percentage reductions referring to numerical values.

In this case, with the assigned constraints, the solution of the optimization problem has been found for the combination:

$$S = 192.95 \text{ mm}, T1 = 0.0454 \text{ s}, TT = 0.11 \text{ s}$$

Table 6 shows the numerical and regression models' values correlation, for the blank0 plan, and relative error evaluation. Compared to the original plan, an additional line has been added to show the run values for which it is possible to reach the feasibility limit for the output on the maximum percentage thinning.

**Table 6.** Numerical and regression model values correlation for the blank0 plan.

| Run | Input | | | Regression Model Outputs | | | Numerical Outputs | | | Errors | | |
|---|---|---|---|---|---|---|---|---|---|---|---|---|
| | S (mm) | T1 (s) | TT (s) | Max Thick (%) | Min Thick (%) | Max Rforce DIE (N) | Max Thick (%) | Min Thick (%) | Max Rforce DIE (N) | Max Thick (%) | Min Thick (%) | Max Rforce DIE (N) |
| 1 | 50.00 | 0.0748 | 0.1874 | 20.02 | −7.34 | 2093295 | 20.20 | −7.50 | 2022870 | −0.91 | −2.10 | 3.48 |
| 2 | 99.28 | 0.0916 | 0.1729 | 20.78 | −7.17 | 2182846 | 20.00 | −7.30 | 1918110 | 3.92 | −1.83 | 13.80 |
| 3 | 211.92 | 0.0944 | 0.1439 | 22.11 | −6.92 | 1993481 | 22.10 | −6.90 | 2013660 | 0.04 | 0.30 | −1.00 |
| 4 | 57.04 | 0.0468 | 0.1826 | 18.80 | −7.58 | 1892017 | 19.30 | −7.40 | 1968800 | −2.60 | 2.48 | −3.90 |
| 5 | 226.00 | 0.0832 | 0.2020 | 22.40 | −6.99 | 2063007 | 22.10 | −7.10 | 1989740 | 1.35 | −1.57 | 3.68 |
| 6 | 134.48 | 0.0440 | 0.2165 | 21.22 | −7.05 | 1897945 | 21.00 | −7.00 | 2008330 | 1.05 | 0.65 | −5.50 |
| 7 | 197.84 | 0.0356 | 0.2116 | 25.47 | −6.19 | 2109164 | 25.30 | −6.30 | 1999097 | 0.65 | −1.76 | 5.51 |
| 8 | 78.16 | 0.0860 | 0.1245 | 22.47 | −6.78 | 2547603 | 23.10 | −6.60 | 2755880 | 0.31 | 1.22 | −6.73 |
| 9 | 204.88 | 0.0384 | 0.1632 | 24.86 | −6.29 | 1936617 | 25.00 | −6.20 | 1982690 | −0.55 | 1.41 | −2.32 |
| 10 | 218.96 | 0.0664 | 0.1487 | 23.37 | −6.65 | 2011086 | 22.60 | −6.70 | 2008650 | 3.43 | −0.68 | 0.12 |
| 11 | 92.24 | 0.0888 | 0.2213 | 20.00 | −7.29 | 2081651 | 20.00 | −7.40 | 2039430 | 0.00 | −1.43 | 2.07 |
| 12 | 176.72 | 0.0720 | 0.2310 | 21.56 | −7.07 | 2061384 | 21.60 | −7.00 | 2023390 | −0.17 | 0.94 | 1.88 |
| 13 | 120.40 | 0.0692 | 0.1971 | 19.88 | −7.37 | 1888758 | 19.60 | −7.30 | 1981650 | 1.44 | 0.99 | −4.69 |
| 14 | 162.64 | 0.0804 | 0.1681 | 20.83 | −7.20 | 1915638 | 21.10 | −7.00 | 1927210 | −1.27 | 2.79 | −0.60 |
| 15 | 155.60 | 0.0972 | 0.2068 | 20.32 | −7.31 | 1696835 | 20.70 | −7.20 | 1893290 | −1.85 | 1.51 | −10.38 |
| 16 | 183.76 | 0.0496 | 0.1148 | 23.38 | −6.51 | 1556814 | 23.50 | −6.70 | 1200260 | 4.85 | −5.71 | −17.74 |
| 17 | 141.52 | 0.1000 | 0.1390 | 21.95 | −6.89 | 2071463 | 21.80 | −6.90 | 2001060 | 0.70 | −0.20 | 3.52 |
| 18 | 85.20 | 0.0636 | 0.1536 | 20.00 | −7.34 | 2177074 | 19.30 | −7.40 | 2130240 | 3.60 | −0.81 | 2.20 |
| 19 | 64.08 | 0.0412 | 0.1342 | 19.57 | −7.41 | 2086636 | 19.70 | −7.40 | 2100820 | −0.67 | 0.16 | −0.68 |
| 20 | 190.80 | 0.0608 | 0.1923 | 22.36 | −6.90 | 1770676 | 23.30 | −6.90 | 1200110 | 0.29 | 0.02 | −11.90 |
| 21 | 169.68 | 0.0776 | 0.1197 | 22.20 | −6.81 | 1906247 | 22.80 | −6.90 | 1977980 | −2.62 | −1.34 | −3.63 |
| 22 | 106.32 | 0.0580 | 0.1100 | 21.37 | −6.97 | 1899818 | 20.60 | −7.00 | 1873870 | 3.74 | −0.39 | 1.38 |
| 23 | 127.44 | 0.0328 | 0.1294 | 21.40 | −6.94 | 1947060 | 21.60 | −7.00 | 1889500 | −0.93 | −0.84 | 3.05 |
| 24 | 148.56 | 0.0524 | 0.1584 | 21.15 | −7.08 | 1807338 | 21.50 | −7.00 | 2045660 | −1.63 | 1.15 | −11.65 |
| 25 | 71.12 | 0.0552 | 0.2262 | 19.18 | −7.44 | 2027902 | 19.10 | −7.40 | 2024410 | 0.39 | 0.59 | 0.17 |
| 26 | 113.36 | 0.0300 | 0.1778 | 20.59 | −7.15 | 2059791 | 20.30 | −7.30 | 1952500 | 1.45 | −2.04 | 5.50 |
| Min | 50.00 | 0.030 | 0.11 | 18.80 | −7.58 | 1556814 | 19.10 | −7.50 | 1200110 | −2.62 | −5.71 | −17.74 |
| Max | 226.00 | 0.10 | 0.23 | 25.47 | −6.19 | 2547603 | 25.30 | −6.20 | 2755880 | 4.85 | 2.79 | 13.80 |

In this case, with the assigned constraints, the solution of the optimization problem has been found for the combination:

$$S = 192.95 \text{ mm}, T1 = 0.0454 \text{ s}, TT = 0.11 \text{ s}$$

The expected results of the model are reported in Table 7.

**Table 7.** Optimal combination of regression model; Blank0.

| S (mm) | T1 (s) | TT (s) | Max Rforce DIE (N) | Max Thick (%) | Min Thick (%) | Objective and Penalty | Objective Function (N) | Penalty Function |
|---|---|---|---|---|---|---|---|---|
| 182.847974 | 0.0454548 | 0.11000185 | 1472516 | 20 | −6.42 | 1472516 | 1472516 | 0 |

The Pearson correlation or linear correlation (a measure of the strength of the association between the two variables) is calculated as follows for the X and Y parameters:

$$r_{xy} = \frac{\sum_{k=1}^{N} (x_k - \overline{x})(y_k - \overline{y})}{\sqrt{\sum_{k=1}^{N} (x_k - \overline{x})^2} \sqrt{\sum_{k=1}^{N} (y_k - \overline{y})^2}} \tag{1}$$

Where:

- $k$ is the sample size
- $x_k$, $y_k$ are the individual sample points indexed with $k$
- $\overline{x} = \frac{1}{n} \sum_{k=1}^{n} x_k$  $\overline{y} = \frac{1}{n} \sum_{k=1}^{n} y_k$

The "$r$" values will be between the range −1 and 1, where the first value represents a perfect inverse linear correlation, and the second a perfect direct linear correlation. The values close to zero and zero itself are indicative of a poor parameters correlation. Figure 24 shows the linear correlation matrix for the blank0 plan.

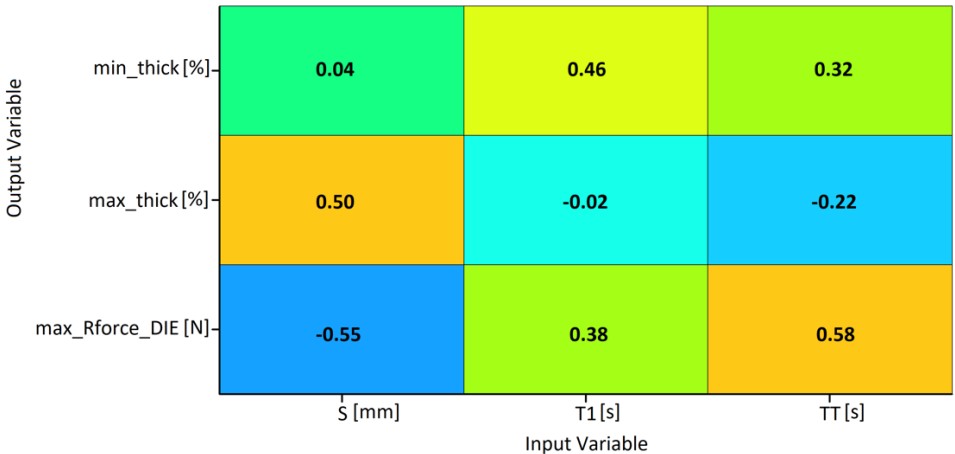

**Figure 24.** Linear correlation matrix for the blank0 plan.

Figure 24 shows no correlation between the space before the motion inversion (S) and the minimum percentage thinning. The same result is seen between the time on the first motion inversion (T1) and the maximum percentage-thinning. In contrast, the maximum reaction force and the process end time (TT) have a very strong direct correlation, whereas the space S and the maximum reaction force are, instead, inversely correlated.

Figure 25 shows a history related to the research for the solution by the algorithm: the red dots correspond to solutions that were unable to satisfy the given constraints, the black dots the solutions able to satisfy them, and the green dots (located in X and Y with the pink-colored axes) represent the optimal solution in the explored design space.

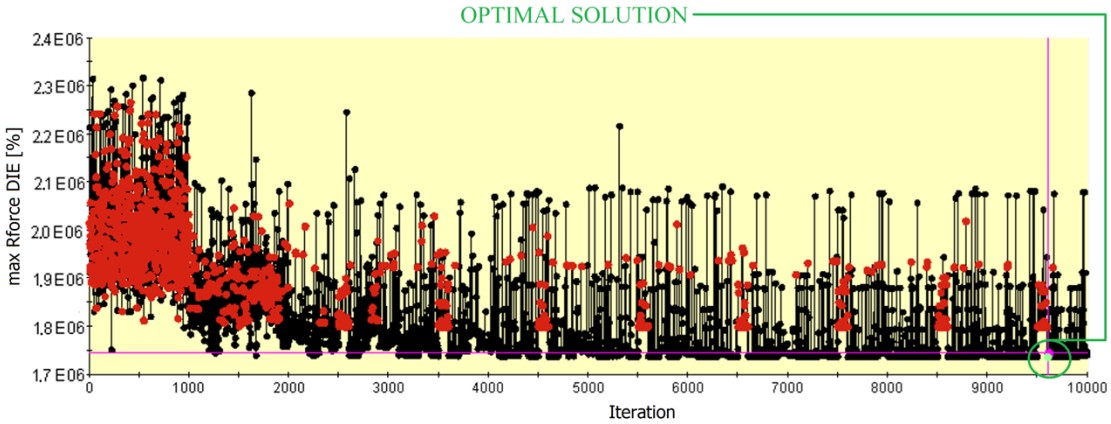

**Figure 25.** Design explored with multi-island genetic algorithm (MIGA), blank0 plan.

### 3.3. Definition and Resolution of the Optimization Problem for Blank1

As in the case of blank0, below is reported the optimization problem definition for the model blank1 plan. The optimization problem has been defined as follows:

| | |
|---|---|
| Objective Function: | Min (max_Rforce_Die) |
| Design Variables: | 50 mm ≤ S ≤ 226 mm, |
| | 0.03 s ≤ T1 ≤ 0.1 s, |
| | 0.11 s ≤ TT ≤ 0.231 s |
| Constraints: | 19% ≤ max_thick ≤ 25%, −9% ≤ min_thick ≤ −6% |

The design space constraints have been assigned by the maximum and minimum values of percentage reductions referring to numerical values. Table 8 shows the numerical and regression model values correlation, for the blank1 plan, and relative error. Compared to the original plan, an additional line has been added to show the run values for which it is possible to reach the feasibility limit for the output on the maximum percentage thinning.

**Table 8.** Numerical and regression model values correlation for blank1 plan.

| | Input | | | Regression Model Outputs | | | Numerical Outputs | | | Errors | | | |
|---|---|---|---|---|---|---|---|---|---|---|---|---|---|
| Run | S (mm) | T1 (s) | TT (s) | Max Thick (%) | Min Thick (%) | S (mm) | T1 (s) | TT (s) | Max Thick (%) | Max Thick (%) | S (mm) | T1 (s) | |
| 1 | 50.00 | 0.0748 | 0.1874 | 19.80 | −7.42 | 1834375.021 | 20.20 | −7.50 | 1835260 | −1.96 | −1.07 | −0.05 | |
| 2 | 99.28 | 0.0916 | 0.1729 | 20.12 | −7.49 | 1937125.107 | 19.50 | −7.60 | 1964070 | 3.17 | −1.51 | −1.37 | |
| 3 | 211.92 | 0.0944 | 0.1439 | 22.17 | −7.02 | 1972896.543 | 21.60 | −7.00 | 1981810 | 2.66 | 0.23 | −0.45 | |
| 4 | 57.04 | 0.0468 | 0.1826 | 19.05 | −7.58 | 1867188.059 | 19.10 | −7.30 | 1871000 | −0.25 | 3.82 | −0.20 | |
| 5 | 226.00 | 0.0832 | 0.2020 | 22.41 | −7.27 | 2311269.029 | 21.90 | −7.30 | 2371100 | 2.32 | −0.47 | −2.52 | |
| 6 | 134.48 | 0.0440 | 0.2165 | 20.84 | −7.61 | 2019189.077 | 21.90 | −8.20 | 2045330 | −4.84 | −7.16 | −1.28 | |
| 7 | 197.84 | 0.0356 | 0.2116 | 24.39 | −7.57 | 2158267.881 | 24.00 | −7.50 | 2168100 | 1.62 | 0.92 | −0.45 | |
| 8 | 78.16 | 0.0860 | 0.1245 | 22.79 | −6.72 | 1880775.054 | 22.80 | −6.60 | 1877480 | −0.05 | 1.86 | 0.18 | |
| 9 | 204.88 | 0.0384 | 0.1632 | 23.84 | −8.09 | 2138040.466 | 24.30 | −7.20 | 2178410 | −1.91 | 12.37 | −1.85 | |
| 10 | 218.96 | 0.0664 | 0.1487 | 23.37 | −7.60 | 2146448.837 | 24.20 | −7.70 | 2060450 | −3.41 | −1.26 | 4.17 | |
| 11 | 92.24 | 0.0888 | 0.2213 | 19.26 | −7.45 | 1965398.706 | 19.10 | −7.50 | 1939120 | 0.83 | −0.60 | 1.36 | |
| 12 | 176.72 | 0.0720 | 0.2310 | 21.55 | −7.28 | 2057738.866 | 21.90 | −7.20 | 2044910 | −1.61 | 1.05 | 0.63 | |
| 13 | 120.40 | 0.0692 | 0.1971 | 19.69 | −7.71 | 2028697.012 | 19.40 | −7.40 | 1858140 | 1.47 | 4.17 | 9.18 | |
| 14 | 162.64 | 0.0804 | 0.1681 | 20.66 | −7.56 | 1956293.479 | 21.20 | −7.00 | 2001820 | −2.52 | 7.94 | −2.27 | |
| 15 | 155.60 | 0.0972 | 0.2068 | 19.58 | −7.31 | 2022827.292 | 20.10 | −7.30 | 2089730 | −2.59 | 0.19 | −3.20 | |
| 16 | 183.76 | 0.0496 | 0.1148 | 23.54 | −7.52 | 2057717.659 | 23.40 | −7.50 | 2076100 | 0.62 | 0.32 | −0.89 | |
| 17 | 141.52 | 0.1000 | 0.1390 | 21.55 | −6.69 | 1841041.399 | 22.30 | −7.00 | 1753830 | −3.38 | −0.62 | 4.97 | |

**Table 8.** *Cont.*

| Run | Input | | | Regression Model Outputs | | | | | Numerical Outputs | | | | Errors | |
|---|---|---|---|---|---|---|---|---|---|---|---|---|---|---|
| | S (mm) | T1 (s) | TT (s) | Max Thick (%) | Min Thick (%) | S (mm) | T1 (s) | TT (s) | Max Thick (%) | Max Thick (%) | | | S (mm) | T1 (s) |
| 18 | 85.20 | 0.0636 | 0.1536 | 20.29 | −7.66 | 1942531.283 | 19.80 | −7.60 | 2058790 | 2.50 | | | 0.78 | −5.65 |
| 19 | 64.08 | 0.0412 | 0.1342 | 20.31 | −7.48 | 1895316.719 | 21.20 | −7.60 | 1824710 | −4.18 | | | −1.62 | 3.87 |
| 20 | 190.80 | 0.0608 | 0.1923 | 22.08 | −7.70 | 2140322.430 | 22.00 | −7.80 | 2020780 | 0.38 | | | −1.34 | 5.92 |
| 21 | 169.68 | 0.0776 | 0.1197 | 22.72 | −6.94 | 1910920.430 | 22.70 | −6.80 | 1949350 | 0.09 | | | 2.11 | −1.97 |
| 22 | 106.32 | 0.0580 | 0.1100 | 22.36 | −6.99 | 2048391.365 | 21.90 | −7.20 | 2069810 | 2.11 | | | −2.91 | −1.03 |
| 23 | 127.44 | 0.0328 | 0.1294 | 21.23 | −8.02 | 1960883.548 | 21.20 | −7.80 | 1908440 | 0.12 | | | 2.85 | 2.75 |
| 24 | 148.56 | 0.0524 | 0.1584 | 20.99 | −7.96 | 1997574.579 | 20.90 | −8.10 | 2098800 | 0.45 | | | −1.78 | −4.82 |
| 25 | 71.12 | 0.0552 | 0.2262 | 19.27 | −7.45 | 1880469.975 | 19.10 | −7.60 | 1911420 | 0.88 | | | −1.97 | −1.62 |
| 26 | 113.36 | 0.0300 | 0.1778 | 19.96 | −7.91 | 1935405.070 | 19.30 | −8.10 | 1948510 | 3.40 | | | −2.32 | −0.67 |
| Min | 50.00 | 0.030 | 0.11 | 19.05 | −8.09 | 1834375.021 | 19.10 | −8.20 | 1753830 | −4.84 | | | −7.16 | −5.65 |
| Max | 226.00 | 0.10 | 0.23 | 24.39 | −6.69 | 2311269.029 | 24.30 | −6.60 | 2371100 | 3.40 | | | 12.37 | 9.18 |

In this case, with the assigned constraints, the solution of the optimization problem has been for the combination:

$$S = 64.24 \text{ mm, } T1 = 0.0318 \text{ s, } TT = 0.131 \text{ s}$$

The expected results of the model are reported in Table 9.

**Table 9.** Optimal combination of regression model for Blank1.

| S (mm) | T1 (s) | TT (s) | Max Rforce DIE (N) | Max Thick (%) | Min Thick (%) | Objective and Penalty | Objective Function (N) | Penalty Function |
|---|---|---|---|---|---|---|---|---|
| 64.23899 | 0.031864 | 0.131 | 1863534.957 | 20 | −7.43 | 1863534.957 | 1863534.957 | 0 |

From the correlation matrix shown in Figure 26, it can be seen that output max_Rforce_DIE and max_thick do not depend on the end of process times (TT), in the same way the min_thick does not depend on S. Instead, max_Rforce_DIE and the space S are directly related, the latter is inversely related to max_thick. The search history of the solution is reported in Figure 27.

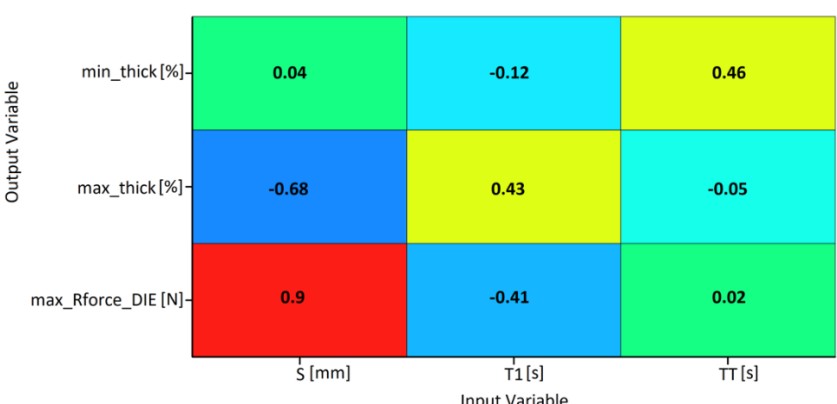

**Figure 26.** Linear correlation matrix, blank 0.

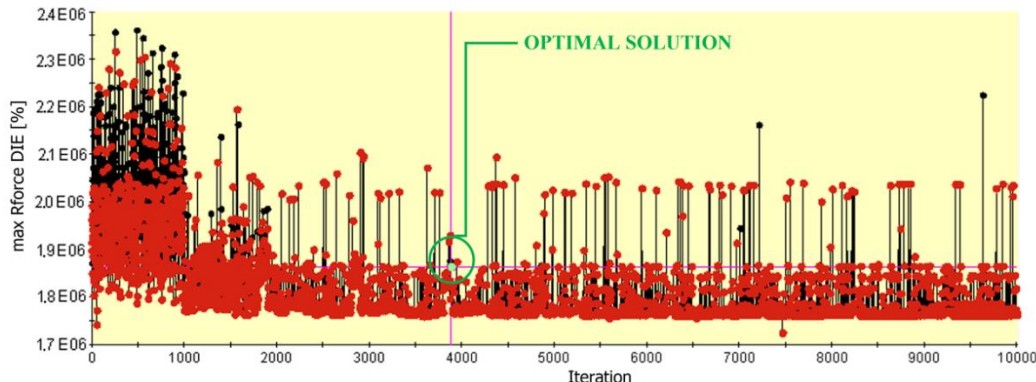

**Figure 27.** Design explored with MIGA for the blank1 plan.

## 4. Model Validation

In order to test the optimization model reliability, numerical analysis runs were performed for the combinations as reported in Tables 7 and 9, in accordance with parameter combinations relating to the optimal solution suggested by the optimization procedure. The punch-displacement input curve definition is reported for the two "optimal runs" in Figure 28. The results for the two runs are shown below. In particular, Figure 29 reports the trend of die reaction force curves, both for the blank0 and blank1 cases. It is evident how, when a stepwise happens, it is possible to detect a reaction force reduction (Figures 28 and 29). For both cases there is a slope variation of the reaction force in the coining phase.

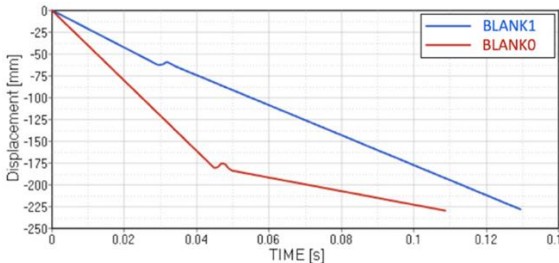

**Figure 28.** Displacement vs time comparison for the optimal combination (according to the regression model) for blank0 and blank1 plan.

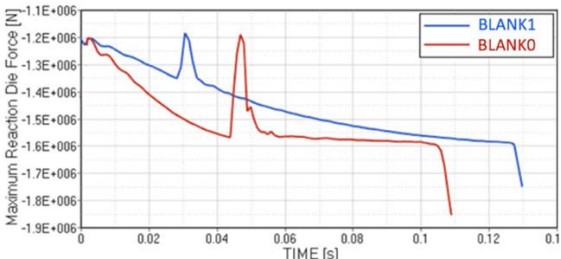

**Figure 29.** Die reaction force comparison for both simulation plans.

Figures 30 and 31 report the percentage-thinning distribution (max and min) and X1, X2, Y1 and Y2 outputs for the optimal combination relative to the blank0 and blank1.

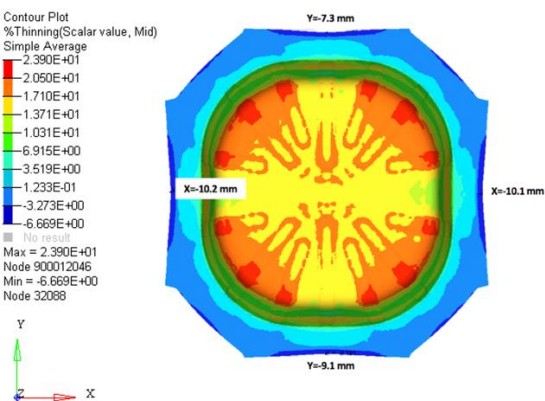

**Figure 30.** %Thinning distribution and X1, X2, Y1 and Y2 for optimal combination for blank0.

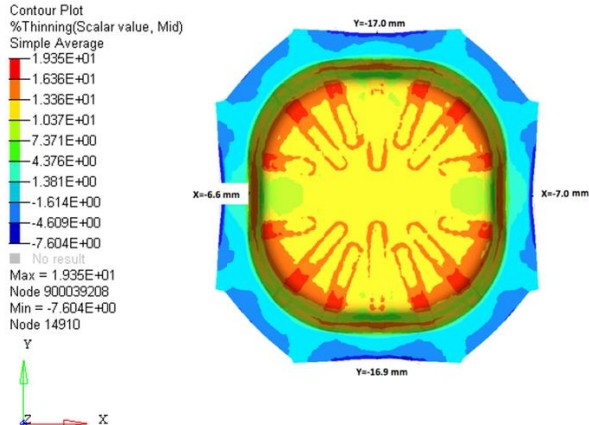

**Figure 31.** %Thinning distribution and X1, X2, Y1 and Y2 for optimal combination for blank1.

Thinning-percentage distribution shows very low values for the blank1, while it remains similar for the runs performed with the plan for the blank0. This result shows, in addition, that the blank reduction involves a thinning reduction, as well as a scrap reduction, without compromising the obtaining of the final geometry of the part. The output values X1, X2, Y1 and Y2 are all negative because they exceed the draw bead profile (on the 4 sides) in the recall direction of material in the die.

Table 10 reports the outputs for the optimal models (blank0 and blank1).

**Table 10.** Output for optimal models: blank0 (optimal0) and blank 1 (optimal1).

| Run | Max Thickness (%) | Min Thickness (%) | Max Rforce DIE (N) | X1 (mm) | X2 (mm) | Y1 (mm) | Y2 (mm) |
|---|---|---|---|---|---|---|---|
| Optimal0 | 23.9 | −6.7 | 1855 | −10.2 | −10.1 | −9.1 | −7.3 |
| Optimal1 | 19.3 | −7.6 | 1751 | −6.6 | −7.0 | −16.9 | −17.0 |

To evaluate the model performance, in Table 11 the percentage error calculation of numerical results respect to regression model results is reported.

**Table 11.** Percentage error calculation of numerical results vs regression model results.

| Run | Max Thick Error (%) | Min Thick Error (%) | Max Rforce DIE Error (kN) |
|---|---|---|---|
| Optimal0 | −19.5 | 4.4 | −26% |
| Optimal1 | 3.2 | 2.3 | 6.0% |

As can be seen from Tables 10 and 11, the model for blank1 conditions appears to be more reliable than the model obtained for blank0 conditions.

## 5. Conclusions

The reported research activity demonstrates how it is possible to support the servo press adoption in industrial contexts with appropriate innovative optimization procedures in order to maximize the positive effect given by their application. In fact, the proposed optimization procedure allows the manufacturing engineers to explore the best servo press configurations for any given process combination in terms of material, thickness and geometry of the formed component. The obtained results have provided an example of the effectiveness of the proposed approach. The authors have to proceed with additional specific research to increase the robustness of the proposed methodologies. In fact, it is important to consolidate what has been developed in connection to a useful correlation with experimental activity in which several process combinations are investigated (material, initial thickness, geometry of the formed component) in order to evaluate the influence of the possible combination on the reliability of the proposed approach. Another element of development is represented by the possible adoption of different optimization algorithms in order to evaluate the optimization results sensitivity to the proposed optimization strategy.

**Author Contributions:** Both A.D.P. and T.P. have defined the proposed methodology. Specifically, A.D.P. has studied the implementation of the optimization model; T.P. has performed numerical simulations and experimental-numerical correlation. Both have contributed to a critical analysis of the obtained results. All authors have read and agreed to the published version of the manuscript.

**Funding:** This research received no external funding.

**Acknowledgments:** The authors thank the company Patrone and Mongiello for the opportunity.

**Conflicts of Interest:** The authors declare no conflict of interest.

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
