# Peer review of "Sheet Metal Forming Optimization Methodology for Servo Press Process Control Improvement"

_metals, doi:10.3390/met10020271_

Round 1

Reviewer 1 Report

Application of servo presses is certainly a qualitative change in the metal forming production practice. Therefore taking up this topic is very pleasing and the presented results are interesting. However, the article in this form does not satisfy all scientific paper criteria.

GENERAL REMARKS

Despite the fact, that sheet metal forming modeling is the main tool, and especially because, that it is the main source of data for further optimization analysis, there is no exact desription of the numerical model. Even very basic components of such a numerical procedure is not presented. For example, what method was used for metal forming analysis? FEM? FVM? Other?Assuming FEM, was any model optimization procedure applied before results were verified with real tests? What type of model was applied: static or dynamic? What types of elements and remeshing procedure were used? What types of tools applied - rigid or deformable? etc.

Through the article, the authors used several times the term "material recall". The context implies that it is about a "springback" which is formal name for the sheet material behavior after forming. Is it?

DETAILED remarks

41-53

It is not clear, is it a list o f sero-press advantages? Is that list a summary by authors based on previously mentioned literature or sth else?

55-70

At the end of the introduction, the authors describe wrinkling as one of the crucial defects in stamping, which is really true, but there are other types of drawpieces defects, too. There is no explanation, why wrinkling is the main point of interest or why authors concentrate on it mainly, at this point. There is no particular reference to wrinkling later in the article.

111

On what basis the coefficient of friction was assumed?

Figure 3

Meaning of the red and blue lines is not precisely described. Thus, X1, X2, Y1, and Y2 measurement, and following that further results analysis - tables 4,5 - are not clear. Drawing of the blank before forming process over the drawbead might be helpful.

Table 2 - consider placing it closer to figure 3 in the text for better understanding.

124

"The output variables set is shown in Table 1"

Should be "Table 2"

131

"In particular, the material changes its performance based on the deformation history"

Disputable. Consider PROPERTIES instead of PERFORMANCE.

The same line

"This property is the main feature ..."

Performance is not a property!

Fig 5-6

What is rigid body, mentioned in the upper right corner?

149-154

Very unclear. What basis these parameters were assumed on? Earlier experiments? Numerical analysis? Literature survey?

Fig. 7

A blank geometry description should be palced at the beginning of the numerical methodology description, since it was used also for numerical analysis of sheet forming, right? And there was no information abour the blank geometry at that place, while - the authors admitt that - it is important for the forming process.

166

"The size reduction in the perpendicular direction to the blank feed, derives from the possibility of always being able to use the same tool for shearing"

OK, but why authors decrease both sides exactly by 7,5mm?

Figure 11

Description not clear. Displacement corresponds to S and T1?

GRAMMAR/LANGUAGE

Excesive use of expression THANKS TO. Try to use other forms, through, in connection, in conjuction owing to , ...

It seems, that in many places in the article one may find PLANE instead of PLAN!

31

"...flexibiity of a hydraulic press (infinite sliding (ram) speed and position control, available of press...)"

Instead of double brackets consider dash: (infinite sliding - ram - speed and...)

55-70

All is about wrinkling so making an indentation almost every sentence seems strange and inappropriate. It looks like a one paragraph.

57

"On the contrary an high blank holder ..."

consider rather with comma:

On the contrary, an high black holder ...

73

"...improve quality ..."

improve PRODUCT quality sounds better

97

"...obtained thanks to a Cx2 processing method ..."

obtained in (or by, or through) a Cx2 ...

101

"This run will be indicated as RUN0..."

be careful with the tense; it may suggest that it will be your future analysis, not already done.

141

Not elegant reference to figures

155

"... variables depending by ..."

rather : depending ON

344

"...reliable than the model obtained for blank0 ones"

rather: blank0 ONE

Reviewer 2 Report

line 100 - units mm/s,

line 111- table 1- explain the parameters, commas for periods,

line 140 - fig. 6 – poor image quality,

line 157 - table 3 unit s,

line 174 - table 4 - dots instead of comma ,

line 179 - table 5 - dots instead of comma,

Fig. 8-15 complete the part- discussion of the results,

line 133- fig. 16 - poor quality - numbers too small, similarly to fig. 17,19, 22 units?

Note to all figures:  the numbers on the axes - the correct notation: dots instead of comma,

line 248-249 - dots instead of comma, editing errors.

line 257- table 6 - incorrect numeric entry (.instead of ,)

line 265- table 7 - incorrect numeric entry (. instead of ,)

line 267 - formula 1- explain the parameters in the formula

line 271- fig. 23 - incorrect numeric entry,

line 289-291- necessary editing,

line 297- table 8 - incorrect numeric entry,

line 305- table 9 - incorrect numeric entry,

Figs. 24 and 26 - no description of the x axis, y axis - no unit

Fig. 27-30 - complete the results discussion.

General remark - the discussion of the results presented in the article is too small, it is necessary to supplement.

Literature - should be improved as required by the Journal Metals

Reviewer 3 Report

Some figures require improvement of the quality. It is known, that the figures presented in the manuscript are generated directly from the program, as the result of the post-processing. However their quality should be improved, because in its current form they are unreadable and therefore difficult to understanding and interpret:
Figures: 1,2,8,912,13,29,30 - the legend is unreadable, the font is too small.
Figures: 16-22,25 - the axis description is unreadable, the font is too small.
Figure 24 - the axis description is difficult to interpret:
1400000 = 14 E6 ....e.t.c

Reviewer 4 Report

Presented research is interesting but contains several formal inaccuracies and technical inconsistencies.

An explanation of abbreviations and symbols should be added under the Table 1 and Table 2. The chapters should be numbered. The decimal number should be separated by a decimal point. Lines 248-250; 288-201 are unreadable. Tables 6 and equation (1) are not in correct form. Line 268 - explanation of "r" is needed (Pearson correlation).

Round 2

Reviewer 1 Report

I have just finished reading the article after corrections. I confirm, that the majority of my remarks, especially the main ones, have been taken into account.